# Remnant tissue enhances early postoperative biomechanical strength and infiltration of Scleraxis-positive cells within the grafted tendon in a rat anterior cruciate ligament reconstruction model

Junki Kawakami[1☯], Satoshi Hisanaga[1☯], Yuki Yoshimoto[2,3], Tomoji Mashimo[4], Takehito Kaneko[5], Naoto Yoshimura[1], Masaki Shimada[1], Makoto Tateyama[1], Hideto Matsunaga[1], Yuto Shibata[1], Shuntaro Tanimura[1], Kosei Takata[1], Takahiro Arima[1], Kazuya Maeda[1], Yuko Fukuma[1], Masaru Uragami[1], Katsumasa Ideo[1], Kazuki Sugimoto[1], Ryuji Yonemitsu[1], Kozo Matsushita[1], Masaki Yugami[1], Yusuke Uehara[1], Takayuki Nakamura[1], Takuya Tokunaga[1], Tatsuki Karasugi[1], Takanao Sueyoshi[1], Chisa Shukunami[3], Nobukazu Okamoto[1], Tetsuro Masuda[1]*, Takeshi Miyamoto[1,6]*

1 Faculty of Life Sciences, Department of Orthopaedic Surgery, Kumamoto University, Chuo-ku, Kumamoto, Japan, 2 Department of Molecular Craniofacial Embryology, Graduate School of Medical and Dental Sciences, Tokyo Medical and Dental University, Bunkyo-ku, Tokyo, Japan, 3 Department of Molecular Biology and Biochemistry, Basic Life Sciences, Graduate School of Biomedical and Health Sciences, Minami-ku, Hiroshima, Japan, 4 Division of Animal Genetics, Laboratory Animal Research Center, The Institute of Medical Science, The University of Tokyo, Tokyo, Japan, 5 Graduate School of Science and Engineering, Iwate University, Morioka, Iwate, Japan, 6 Department of Orthopedic Surgery, Keio University School of Medicine, Shinjuku-ku, Tokyo, Japan

☯ These authors contributed equally to this work.
* temasuda@kuh.kumamoto-u.ac.jp (TM); miyamoto.takeshi@kuh.kumamoto-u.ac.jp (TM)

## Abstract

When ruptured, ligaments and tendons have limited self-repair capacity and rarely heal spontaneously. In the knee, the Anterior Cruciate Ligament (ACL) often ruptures during sports activities, causing functional impairment and requiring surgery using tendon grafts. Patients with insufficient time to recover before resuming sports risk re-injury. To develop more effective treatment, it is necessary to define mechanisms underlying ligament repair. For this, animal models can be useful, but mice are too small to create an ACL reconstruction model. Thus, we developed a transgenic rat model using control elements of Scleraxis (Scx), a transcription factor essential for ligament and tendon development, to drive GFP expression in order to localize Scx-expressing cells. As anticipated, Tg rats exhibited Scx-GFP in ACL during developmental but not adult stages. Interestingly, when we transplanted the flexor digitorum longus (FDP) tendon derived from adult Scx-GFP+ rats into WT adults, Scx-GFP was not expressed in transplanted tendons. However, tendons transplanted from adult WT rats into Scx-GFP rats showed upregulated Scx expression in tendon, suggesting that Scx-GFP+ cells are mobilized from tissues outside the tendon. Importantly, at 4 weeks post-surgery, Scx-GFP-expressing cells were more frequent within the grafted tendon when an ACL remnant was preserved (P group) relative to when it was not (R group) (P vs R

**Data Availability Statement:** All relevant data are within the manuscript and its Supporting information files.

**Funding:** No authors received a salary from any of our funders.

**Competing interests:** The authors have declared that no competing interests exist.

groups (both n = 5), p<0.05), and by 6 weeks, biomechanical strength of the transplanted tendon was significantly increased if the remnant was preserved (P vsR groups (both n = 14), p<0.05). Scx-GFP+ cells increased in remnant tissue after surgery, suggesting remnant tissue is a source of Scx+ cells in grafted tendons. We conclude that the novel Scx-GFP Tg rat is useful to monitor emergence of Scx-positive cells, which likely contribute to increased graft strength after ACL reconstruction.

## Introduction

Anterior cruciate ligament (ACL) injuries occur frequently in sports and other activities and often interfere with continuation of sports activities and impair a person's ability to perform activities of daily living [1]. Surgical treatment of ACL injuries has been successful with development of anatomical single- and double-bundle ACL reconstruction [2–4]. However, ACL grafts used in this procedure undergo ligamentization, which includes graft necrosis followed by cellular infiltration and remodeling of the transplant, which may compromise its strength [5–7].) Also, re-ligamentation may not be complete by the time many patients return to sports in some institutions, and the risk of re-tear is reportedly 2–10% at 5 years postoperatively [8].

Various attempts to accelerate the process of tendon reconstruction have been reported, including administration of mesenchymal stem cells, growth factors, or platelet-rich plasma into the operated knee joint during surgery, in either animal or human models [9]. Using an adult sheep ACL reconstruction model, Kondo et al. reported that transplantation of synovial-derived fibroblasts cultured in TGF-β-supplemented medium significantly enhanced infiltration of cells into the transplanted tendon by 3 months postoperatively and significantly increased load-to-failure of the transplanted tendon by 12 weeks postoperatively [10]. However, potential side effects of TGF-β cannot be ruled out, and the high cost of the treatment has not led to its use in actual clinical practice.

Recently, ACL reconstruction with preservation of native ACL remnant tissue has been reported to improve postoperative knee joint stability relative to conventional techniques [11]. The presence of mesenchymal stem cells in remaining tissue, which may facilitate healing of a transplanted tendon, has also been reported [12,13]. ACL-derived vascular stem cells with high differentiation potential are also found in remnant tissues [14], but it remains unclear whether they undergo redifferentiation into ligament cells.

Interestingly, cell lineage tracing analysis has revealed Scleraxis (Scx)-positive cells in ACLs of developing mice [15]. Scx is a basic helix-loop-helix transcription factor expressed in progenitor cells of tendons and ligaments [16,17]. Mice lacking Scx reportedly exhibit relatively decreased expression of Tenomodulin, a marker of tendon and ligament maturation, and Scx-deficient mice show defective tendon and ligament development [18]. However, the function of Scx-positive cells during re-ligamentation of reconstructed ACL is not known.

Here, we hypothesized that remnant tissue might increase the initial strength of a reconstructed ligament and that the appearance of Scx-positive cells could, at least in part, underlie that increase. To test this hypothesis, we newly established Scx-GFP transgenic rats, in which Scx expression can be monitored by GFP expression. Although various genetically engineered mouse models are currently available, including Scx-GFP transgenic mice, it remains quite difficult to create a stable ACL reconstruction model in mice due to their small size. Also, an anti-rat Scx antibody useful for immunohistological analysis has not been available. Thus, in this study we used Scx-GFP transgenic rats to determine whether Scx-expressing cells in remnant

tissue contribute to mechanical strength and histological changes seen in the transplanted tendon after ACL reconstruction surgery.

## Materials and methods

### Animals

Wild-type (WT) Sprague-Dawley rats were purchased from Japan SLC Co Ltd (Hamamatsu, Shizuoka, Japan). The Scx-GFP transgenic construct, in which a GFP sequence was knocked into exon1 of mouse *Scx* gene containing 4kb upstream and 5kb downstream regions of exon1 was injected into pronuclei of fertilized eggs, and Scx-GFP transgenic rats (Scx-GFP Tg rats) were established. Detection of specific GFP expression in the developing ACL in 1-day-old Tg rats confirmed successful establishment of Scx-GFP Tg animals (S1 Fig).

### Rat surgical procedures

Using an anesthesia box, rats were anesthetized by inhalation of isoflurane and intraperitoneal administration of a triad of anesthetics (medetomidine hydrochloride 0.375 mg/kg + midazolam 2.0 mg/kg + butorphanol tartrate 2.5 mg/kg). We then created animal models of ACL reconstruction for remnant preservation models (Group P) and remnant resection models (Group R) in 11- to 12-week-old WT or Scx-GFP Tg male rats (body weight, 424.2 ± 17.4 g).

For the ACL reconstruction and ACL tear models, a parapatellar skin incision was made on the medial side of the knee joint of 11- to 12-week-old WT or Scx-GFP Tg male rats (S2 Fig). A medial parapatellar approach was used to reach the joint cavity, and the patella was dislocated laterally with the lower limb extended. The ACL was identified, and then a tear was created by cutting the central ACL portion (S2 Fig). The knee joint was then washed with saline solution, and the joint capsule and skin were sutured with a 4–0 nylon suture. As for the ACL reconstruction model, a longitudinal incision was made on the medial aspect of the distal leg and ankle after dissecting the ACL (S2 Fig). Approximately 20 mm of the ipsilateral flexor digitorum longus tendon was harvested, and both ends of the harvested flexor digitorum longus tendon was sutured by a 4–0 nylon suture to create a graft. An 18-gauge needle and drill (outer diameter, 1.27 mm) were used to create bone tunnels near the anatomic location of the ACL on the tibial and femoral lateral condyles. For both P and R Groups, one end of a 4–0 nylon suture fixed to the grafted tendon was passed through an 18-gauge needle inserted into bone tunnels, and the grafted tendon was inserted into the tunnel, first in femur and then in the tibia. Both ends of the grafted tendon were fixed with 4–0 nylon sutures to the surrounding periosteum at the distal femur and proximal tibia extra-articular tunnel exit sites. The knee joint was placed in extension, and the grafted tendon was fixed such that there was no slack in the tendon. Postoperative rats were allowed to live freely without behavioral restrictions. For histological or mechanical evaluation, we used 11- to 12-week-old Scx-GFP Tg rats or WT rats (S2 Fig).

### Animal ethics

When rats were euthanized, the procedure was performed humanely, taking into consideration the alleviation of suffering. Using an anesthesia box, rats were anesthetized by inhalation of isoflurane and intraperitoneal administration of a triad of anesthetics (medetomidine hydrochloride 0.375 mg/kg + midazolam 2.0 mg/kg + butorphanol tartrate 2.5 mg/kg). After adequate pain relief, rats were euthanized by cervical dislocation. All animal experiments were carried out in accordance with the Institutional Guidelines on Animal Experiment at Kumamoto University, and animal experiment procedures were approved by the Animal Studies

Committee and the Institutional Animal Care and Use Committee at Kumamoto University, Japan. This study is reported in accordance with ARRIVE guidelines.

## Mechanical evaluation

At 4, 6 or 8 weeks after ACL reconstruction, knees of the P and R Groups (N = 14 each) were subjected to a tensile tear test using a tension measuring device (S3 Fig). For these experiments, we used 11 to 12-week-old wild-type male rats. Their body weights at 3 subsequent biomechanical evaluations were: at week 4 (Group P; 535.6 ± 17.2g, Group R; 512.6 ± 17.1g), at week 6 (Group P; 532.1 ± 22.7g, Group R; 524.4 ± 15.3g), and at week 8 (Group P; 547.1 ± 13.0g, Group R; 541.7 ± 10.8g). Rats were anesthetized in an anesthesia box by isoflurane inhalation and intraperitoneal administration of three anesthetics. Cervical dislocation was then performed, and rats were euthanized. The knee joint was carefully dissected with complete dissection of all soft tissues except the ACL graft. The femur was dislocated from the hip joint, leaving the entire length of the femur and tibia. A 1.2-mm-diameter drill was used to create a bone hole in the center of the femur and tibia, and a 1-mm-diameter mild steel wire was passed through the bone hole so that it could be grasped by a tension measuring device. Nylon sutures used to fix the ACL graft onto both the femur and tibia were removed prior to biomechanical testing. The harvested knee joint was wrapped in gauze soaked in saline solution, stored on ice, and tested for tensile fracture on the day of sampling. The prepared femur-ACL-tibia composite was mounted on a tension measuring device (Shimadzu, EZ-test). Prior to testing, knee tissue specimens were preconditioned by applying 10 cycles of 1 N longitudinal loading. After preconditioning, ACL thickness was measured. Each femur-ACL-tibia composite was then elongated at a rate of 0.25 mm/s until the ACL tore, and the maximum tensile fracture strength was measured.

## Immunohistological evaluation

To detect Scx-expressing cells within the transplanted tendon in the ACL reconstruction model, knee joint tissue was harvested at 4 or 6 weeks post-surgery, and grafted tendons were immunohistochemically stained for GFP (N = 5 each group). In the ACL tear model, knee joint tissues were harvested on the day of surgery and at weeks 1, 2, and 4 postoperatively for GFP immunohistochemistry to evaluate Scx-expressing cells in remnant tissue (N = 4 each group). To determine the origin of Scx-expressing cells, three ACL reconstruction models were created using 11 to 12-week-old Scx-GFP Tg or wild-type male rats (S4 Fig). For the first model we transplanted wild-type rats with grafts from wild-type rats, and in the second, we transplanted Scx-GFP Tg rats with grafts from wild-type rats. For a third model, wild-type rats were transplanted with grafts from Scx-GFP Tg rats. Knee joint tissues were harvested 4 weeks later and analyzed using fluorescent immunohistochemistry. Anesthesia was administered prior to surgery, and reflux fixation was performed with 4% paraformaldehyde. Knee joint tissues were then harvested and immersion-fixed in 4% paraformaldehyde with 20% sucrose for 24 hours. Tissues were freeze-embedded in SCEM-L1 (Section-lab, Hiroshima, Japan) and stored in a -80˚C freezer. Frozen tissue was then sectioned to 5 μm thickness on a Leica CM3050 S cryostat (Leica, Wetzlar, Germany). Sections were then treated 15 min with 0.05% proteinase-K diluted in PBS at 37˚C for antigen activation and then treated 5 min with Triton-X 100 (Nakalai, Kyoto, Japan) diluted in 0.1% in PBS at 25˚C to permeabilize the membrane. Samples were blocked 30 min in 3% bovine serum albumin (BSA) diluted in PBS at 25˚C and incubated overnight with primary antibodies against GFP (1:600 dilution, MBL, Tokyo, Japan) diluted in 1% BSA at 4˚C. Sections were then incubated 120 min with secondary antibodies conjugated with Alexa Fluor 488 (Thermo Fisher Scientific K.K., Tokyo, Japan) at 4˚C.

Sections were contrast-stained with Vectashield mounting medium for fluorescence with DAPI (Vector Laboratories, Burlingame, USA), observed under a fluorescence microscope (BZ-X800, Keyence, Osaka, Japan). Remnant-preserved or -resected ACL reconstruction models (10 rats each) were created in 11 to 12-week-old Scx-GFP Tg rats. Then, at both 4- and 6-week time points postoperatively, we collected grafted tendons from 5 rats each from either preservation or resection groups and prepared frozen sections from those tendons for staining with anti-GFP antibody. GFP signals in one section from each grafted tendon were analyzed by fluorescence microscopy (BZ-X800, Keyence, Osaka, Japan). The ACL tear model consisted of 4 rats per time point, and we analyzed one section from each rat. Images were analyzed at 40x magnification. For the ACL reconstruction model, we observed the parenchymal area outside the bone tunnel of the grafted tendon, and quantified the extent of the Scx-expressing cell area by dividing the GFP-positive area by the area of the grafted tendon parenchyma. For the ACL tear model, we observed the remnant area, and quantified the extent of the Scx-expressing cell area by dividing the GFP-positive area by the area of the remnant.

## RT-PCR analysis

The ACL tear model was created using 11- to 12-week-old WT rats. ACLs were sampled at days 0 and 2 and weeks 1, 2, and 4 postoperatively, placed in sample tubes and immediately frozen in liquid nitrogen. Subsequently, samples were homogenized using a Multi-bead Shocker (Yasui-Kikai, Osaka, Japan), and total RNA was isolated with QIAzol Lysis Reagent (Qiagen GmbH, Hilden, Germany) and extracted using an RNeasy mini kit (Qiagen GmbH, Hilden, Germany). Single-strand complementary DNAs (cDNAs) were synthesized using ReverTra Ace® qPCR RT Master Mix with gDNA Remover (TOYOBO, Osaka, Japan) and amplified using the THUNDERBIRD® Next SYBR® qPCR Mix (TOYOBO, Osaka, Japan) in a ViiA™ 7 Real-Time PCR System (Applied Biosystems) for 40 cycles. Each cycle consisted of 5 sec denaturation at 95˚C and 30 sec annealing and/or extension at 60˚C. GAPDH served as an internal control. Primers used for RT-PCR were as follows:

GAPDH-5′; 5′-AGGGCTGCCTTCTCTTGTGAC-3′,

GAPDH-3′; 5′- TGGGTAGAATCATACTGGAACATGTAG -3′,

Scx-5′; 5′-GACCTAAAGAGGCGGCATGA -3′,

Scx-3′; 5′-AGCATGAACACGACAGGGTT -3′.

## Statistical analysis

All numerical data are shown as means ± SD. Statistical analysis of 2 subgroups was performed using Student's t-test or the Mann-Whitney $U$ test. Statistical analysis of 3 or more subgroups was performed using one-way ANOVA, followed by a Tukey–Kramer test to determine significance between groups. (*p < 0.05; **p < 0.01; NS, not significant, throughout the paper).

## Results

### Biomechanical strength in the early postoperative period of ACL reconstruction surgery is greater in the Remnant than the non-Remnant group in rats

A rat ACL reconstruction model was created by dividing 11- to 12-week-old wild-type rats into a remnant preservation group (remnant+, Group P) and a remnant resection group

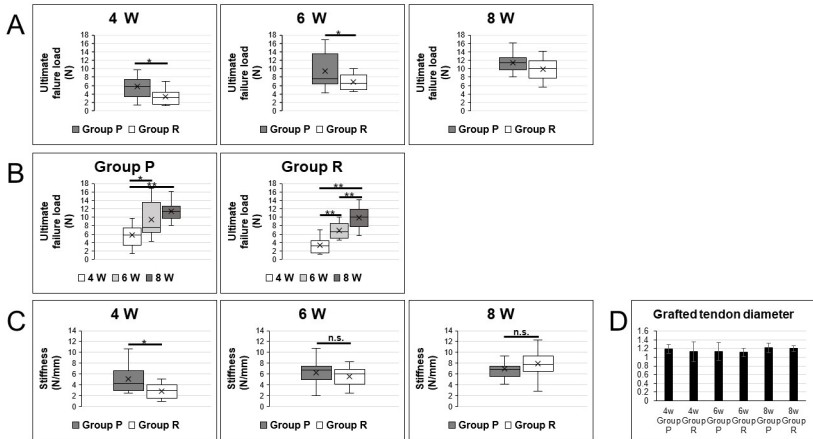

**Fig 1. Biomechanical strength of the grafted tendon is higher in the remnant preservation group compared to the remnant resection group at early postoperative periods.** A rat ACL reconstruction model was created using 11- to 12-week-old wild-type rats divided into two groups: A remnant preservation group (remnant+, Group P) and remnant resection group (remnant-, Group R). Four, 6 or 8 weeks after surgery, knee joint tissue was collected and biomechanical strength was evaluated by assessing ultimate failure load using a tensile strength meter (N = 14 for each group). Ultimate failure load was compared between groups at each time point after surgery (A). Ultimate failure load was also compared in each group among time points after surgery (B). Data represent mean ultimate failure load (N) of the grafted tendon ± SD (n = 14 for each group, *P < 0.05; **P < 0.01). Stiffness was compared between groups at each time point after surgery (C). Data represent mean stiffness (N/mm) of the grafted tendon ± SD (n = 14 per group, *P < 0.05). Grafted tendon diameter did not differ significantly between Groups P and R at any time point (D).

(remnant-, Group R). Knee joint tissues were collected from both groups at 4, 6 or 8 weeks postoperatively. Subsequently, we evaluated biomechanical strength of grafted tendons by ultimate failure load using a tension meter. Grafted tendons ruptured in the center of the parenchyma in all cases, and no tendon was pulled out through the bone hole at any time point after operation. We found that at 4 or 6 weeks postoperatively, the ultimate failure load of grafted tendons was significantly higher in Group P relative to R (*p < 0.05, Fig 1A). However, by 8 weeks postoperatively, we observed no significant difference in ultimate failure load of grafted tendons between the two groups (Fig 1A). Then, within each group, we compared mechanical strength of grafted tendons at 4, 6 and 8 weeks postoperatively. In comparisons made at 4 and 6 weeks or 4 and 8 weeks postoperatively, Group P showed a significant difference in ultimate failure load of grafted tendons (4 weeks vs 6 weeks, *p < 0.05; 4 weeks vs 8 weeks, **p < 0.01; Fig 1B), although those differences were not significant when we compared 6- and 8-week time points. In Group R, ultimate failure load of grafted tendons at 8 weeks postoperatively was significantly higher than at 4 or 6 weeks postoperatively (**p < 0.01, Fig 1B). Similarly, stiffness was significantly greater in Group P relative to R at 4 weeks postoperatively (*p < 0.05, Fig 1C). However, by 6 and 8 weeks postoperatively, we observed no significant difference in stiffness of grafted tendons between groups (Fig 1C). Grafted tendon diameter was not significantly different between Group P and R at any time points (Fig 1D).

## The area of Scx-expressing cells in grafted tendons is significantly larger in remnant+ compared to remnant- groups after ACL reconstruction surgery in Scx-GFP rats

Next, we performed either remnant preservation (remnant+, Group P) or remnant resection (remnant-, Group R) ACL reconstruction surgery in 11- to 12-week-old Scx-GFP rats. At 4 or

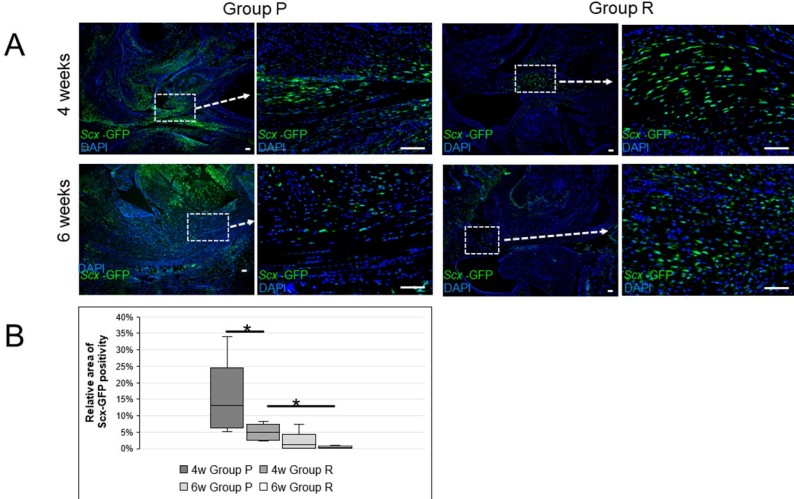

**Fig 2. The remnant preservation group shows increased numbers of Scx-GFP-positive cells in the grafted tendon.**
(A and B) ACL reconstruction surgery was performed in 11- to 12-week-old Scx-GFP Tg rats, which were divided into remnant preservation (remnant+, Group P) and remnant resection (remnant-, Group R) groups. (A) Four or 6 weeks after surgery, knee joint tissue was collected from each group and stained with anti-GFP antibody to detect Scx-expressing cells in transplanted tendon by fluorescent immunohistochemistry. Nuclei were visualized by DAPI. In overviews of parenchyma of transplanted tendon, areas within dashed boxes are shown at higher magnification in the adjacent image. (B) Scx-expressing cells were quantified by analyzing the GFP-positive cell area relative to the total parenchymal area in grafted tendons using a BZ-X700 microscope. Data represent mean GFP-positive cell area relative to total parenchymal area in the grafted tendon ± SD (each with n = 5, *P < 0.05). Magnification is 20x or 100x, respectively, for lower or higher magnification. Bar, 100μm.

6 weeks post-surgery, grafted tendons were collected in each group and evaluated for the presence of Scx-expressing cells in transplanted tendons based on fluorescent immunohistochemistry for GFP signals (Fig 2A). We observed no GFP-positive cells in the flexor digitorum longus tendon prior to transplantation (S5 Fig). However, at 4 or 6 weeks after ACL reconstruction in both groups, we detected GFP-positive cells in transplanted tendons (Fig 2A). We then quantified GFP-positive cell areas in grafted tendons by analyzing the GFP-positive cell area relative to the total parenchymal area in those tendons. The area of GFP-positive cells was significantly greater in grafted tendons dissected from Group P relative to Group R at 4 weeks after surgery (*p < 0.05), but those areas were comparable between groups by 6 weeks postoperatively (Fig 2B). Furthermore, in both groups the GFP-positive cell area in grafted tendons significantly decreased at 6 weeks relative to 4 weeks after surgery (Fig 2B).

## Scx-positive cells are induced in ACL remnant tissue following ACL dissection

Next, we assessed remnant tissues for the presence of Scx-positive cells, given that the Scx-positive cell area in grafted tendons was significantly greater in Group P relative to R at 4 weeks after surgery. To do so, we dissected the ACL in 11- to 12-week-old Scx-GFP rats, collected ACL remnant tissues at specific time points thereafter, and stained them with anti-GFP antibody for immunohistochemical analysis (Fig 3). While Scx-positive cells were not seen in the intact ACL prior to dissection (Fig 3A), they appeared in ACL remnant tissue within 15 minutes of surgery (Fig 3A and 3B). Moreover, in remnant tissues, the number of Scx-positive cells increased gradually until 4 weeks after ACL dissection (*p < 0.05, Fig 3A and 3B). Likewise,

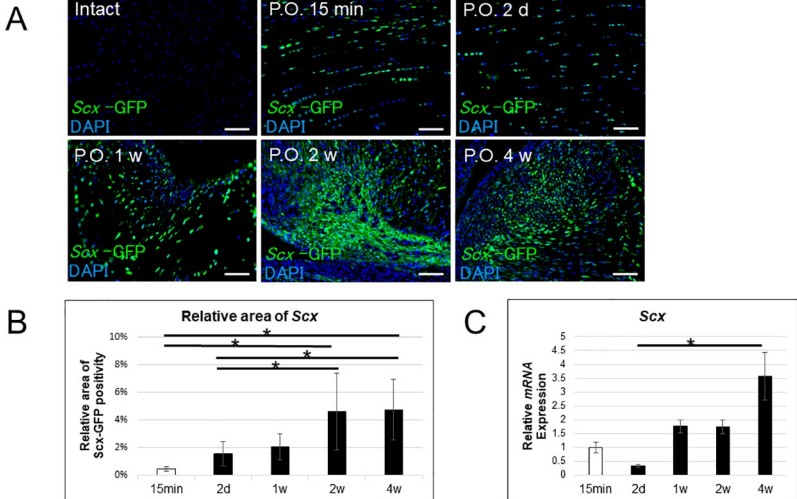

**Fig 3. Induction and increase in number of Scx-positive cells in remnant tissue after ACL dissection.** (A-C) ACLs were dissected in 11- to 12-week-old Scx-GFP Tg rats. (A) Both non-operated (intact) and dissected ACLs were collected at indicated time points, collected tissues were stained with anti-GFP antibody and DAPI, and tissue sections were observed under a fluorescence microscope. Magnification, 100x. Bar, 100μm. (B) Scx-GFP-expressing cells were quantified by analyzing the GFP-positive area relative to the total parenchymal area in grafted tendons. Data represent mean GFP-positive cell area relative to the total parenchymal area in the ACL ± SD (each n = 4, *P < 0.05). Note that samples analyzed on day 0 were collected 15 min post-operatively (P.O.). (C) *Scx* transcript levels were also quantitatively analyzed by realtime PCR in collected ACL tissues. Data represent mean *Scx* relative to *Gapdh* expression in the ACL ± SD (each n = 4, *P < 0.05).

*Scx* transcript levels as determined by quantitative realtime PCR analysis increased and remained high until 4 weeks after ACL dissection in ACL remnant tissues (*p < 0.05, Fig 3C).

### Scx-positive cells within grafted tendons are derived from host tissues

Finally, we evaluated the origin of Scx-expressing cells present within grafted tendons. In the ACL reconstruction model in which either a wild-type or Scx-GFP transgenic rat graft tendon was inserted into a wild-type rat, we observed no GFP-positive cells within the grafted tendon at 4 weeks after surgery (Fig 4, left and right columns). Conversely, in the ACL reconstruction model in which a wild-type graft tendon was inserted into a Scx-GFP transgenic rat, we detected GFP-positive cells in parenchyma within the grafted tendon (Fig 4, middle column). These findings overall indicate that Scx-positive cells within grafted tendons originate from host rather than graft tissues.

## Discussion

ACL injuries are common sports injuries, which affect approximately 200,000 individuals per year in the US [1]. Since the ACL has limited self-repair capacity, surgical treatment is often necessary before a patient can resume sports activities. In fact, 60,000–150,000 individuals undergo surgery annually in the US as treatment for ACL injuries [19]. Autologous tendons like hamstring grafts or bone-patella tendon-bone grafts are widely used for ACL reconstruction surgery. However, due to the time required for the grafted tendon to re-ligamentize and the risk of re-tearing after surgery, a rest period sometimes exceeding 6 months is required before an individual can resume sports activities. Therefore, understanding processes underlying re-ligamentization of the grafted tendon is crucial in order to develop therapies that

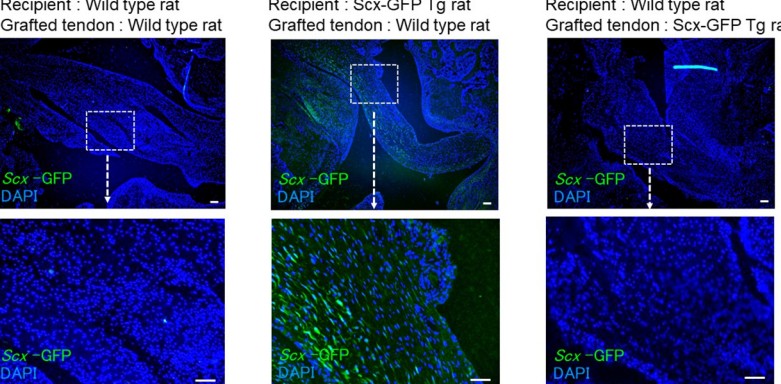

**Fig 4. Scx-positive cells are present in the grafted tendon only after transplant into Scx-GFP Tg rats.** Flexor digitorum longus tendons were collected from 11- to 12-week-old Scx-GFP Tg or wild-type rats for use as grafts. As indicated, wild-type grafts were transplanted into Scx-GFP Tg or wild-type rats, while Scx-GFP Tg grafts were transplanted into wild-type rats. Four weeks later, knee joint tissue was collected stained with anti-GFP antibody to detect Scx-expressing cells by fluorescence microscopy. Top row, lower magnification; Bottom row, higher magnification. Magnification is 20x or 100x, respectively, for lower or higher magnification. Bar, 100μm.

promote proper healing and minimize the rest period following surgery. As tendons repair after ACL reconstruction, cells in the transplant are initially lost due to necrosis but are then replaced by mesenchymal stem cells, allowing re-ligamentization [7]. Administration of growth factors and/or preservation of the ACL remnant reportedly contribute to re-ligamentization and increase strength of the transplanted tendon [10].

Scx is required for embryonic and post-natal growth of ACL and other tendons and ligaments, but its role in tendon repair in adults is unknown. Here, we established Scx-GFP rats and observed that Scx-GFP+ cells, although absent in mature adult tendons, appeared in the graft from outside the tendon during re-ligamentization after ACL reconstruction surgery. This activity was enhanced by the presence of remnant tissue: the number of Scx-GFP+ cells in a grafted tendon in the presence of remnant tissue at 4 weeks postoperatively was significantly higher than in comparable tissue lacking remnant tissue. However, the number of Scx-GFP-positive cells in the transplanted tendon decreased as re-ligamentization proceeded, either with or without remnant tissue, and the number of those cells in both groups was comparable by 6 weeks postoperatively. The early decrease Scx-GFP-positive cell number is anticipated, as Scx-positive cells are not detected in adult tissues, and their loss is thought to indicate re-ligamentization. We have shown that Scx-GFP+ cells seen in ACLs of 1-day-old rats disappear by 12 weeks. In fact, the number of Scx-GFP+ cells in transplanted tendons was significantly higher in the P than the R group at 4 weeks postoperatively, but the biomechanical strength of the transplanted tendons was significantly higher in the P than R group at 6 weeks, indicating a time lag. We identified ACL remnant tissue as a source of Scx-GFP+ cells mobilized to a transplanted tendon, and detected Scx-GFP+ cells in remnant tissue. Both the number of these cells in transplanted tendons and biomechanical strength increased when remnant tissue was retained, suggesting that mobilization of Scx-GFP+ cells into transplanted tendons promotes more rapid re-ligamentization of those tendons and enhances their biomechanical strength.

Creating models for ACL reconstruction surgery in small animals like mice has been technically challenging due to their small size, requiring analysis of a larger animal. However even in larger-sized animals such as rats, visualizing Scx expression has been difficult due to the lack of reliable antibodies to detect Scx protein by immunostaining. In this study, we successfully

established Scx-GFP transgenic rats, enabling visualization of Scx expression for the first time in both remnant tissue and transplanted tendons. Our findings demonstrate that transplanted tendons with a larger number of Scx-positive cells exhibit greater biomechanical strength than those with a lower number of Scx-positive cells.

In mice, Scx expression is reportedly detected in developing tendons and ligaments but absent in adults [15]. However, Scx expression is reportedly induced in adult mice as injured tendons heal [20]. Using Scx-GFP rats, we confirmed Scx-GFP expression in tendons and ligaments of neonatal but not adult rats (Figs 4 and S2), and observed induction of Scx-GFP expression in adult rats during re-ligamentization of grafted tendons (Fig 4). Accordingly, in some tissues, embryonic and developmental programs re-emerge when adult tissues undergo repair [21]. For example, although chondrocytes are required for bone growth during endochondral ossification at embryonic and developmental stages, they disappear in adults [22]; nonetheless, gene expression programs seen during endochondral ossification re-emerge as a fracture heals [23], analogous to re-emergence of Scx-positive cells during tendon and ligament repair. Moreover, we observed a greater number of Scx-GFP+ cells in the P compared to the R group at 4 weeks postoperatively. However, at 6 weeks postoperatively, there was no significant difference in the number of these cells in P and R groups (Fig 2B). In parallel, biomechanical strength was significantly higher in the P compared to the R group at 6 weeks postoperatively, but was comparable in P and R groups by 8 weeks. Although it is not clear why this time lag occurs, our findings strongly suggest that Scx-positive cells are first mobilized into a transplanted tendon, a process enhanced by the presence of remnant tissues. Scx expression likely is downregulated as mobilized cells mature as tenocytes. It is also likely that biomechanical strength of a transplanted tendon in the P group becomes stronger earlier than in the R group due to the rapid mobilization of Scx-positive cells into the transplanted tendon, although the biomechanical strength of each eventually becomes equal. Further analysis of regulation of Scx expression and the activities of Scx-positive cells is needed.

Currently, it remains unclear how remnant tissue promotes infiltration of Scx-positive cells into grafted tendons to enhance biomechanical strength. There are reportedly three phases of re-ligamentation of transplanted tendon after ACL reconstruction: an initial healing phase, the proliferative phase, and a maturation phase [19]. During the proliferative phase, fibroblasts and stem cells from various sources within the knee joint, including the synovial membrane, bone marrow, ACL remnant tissue, and other tissues, infiltrate the transplanted tendon [19,24]. These cells then proliferate within connective tissue and produce collagen and glycosaminoglycans for tissue remodeling during the maturation phase [19]. However, detailed mechanisms relevant to cells involved in re-ligamentation remain unclear. Our study suggests that Scx-positive cells likely function in tissue remodeling within transplanted tendons, and that remnant tissue may stimulate infiltration of these cells into grafted tendons or even serve as a source for these cells. Indeed, Mifune et al. reported an increased number of cell nuclei and greater mechanical strength within the transplanted tendon in a reconstruction model in which a partial ACL tear was augmented with a transplanted tendon at 8 weeks postoperatively [25].

We found that in Scx-GFP transgenic rats, the flexor digitorum longus tendon, which served as the graft, did not exhibit Scx-expressing cells at the time of surgery (S5 Fig). However, when a wild-type rat graft tendon was transplanted into a Scx-GFP transgenic rat using the ACL reconstruction surgery model, Scx-GFP-expressing cells were present within the transplanted tendon (Fig 4, middle panels), indicating that Scx-expressing cells present in grafted tendons are derived from the host.

We also show that *Scx* mRNA expression in remnant tissue gradually increased over a period of 4 weeks postoperatively (Fig 3C). It is reported that mesenchymal stem cells are

present in cells isolated and cultured from human ACL remnant tissue [26] and known that tendon progenitor cells express Scx, and that their differentiation into tendons induces expression of Tnmd and Collagen types 1 and 3 [27]. Thus, in patients, preservation of remnant tissue at the time of ACL reconstruction surgery may increase the number of cells that will eventually differentiate into mature cells expressing type 1 and type 3 collagen.

Our study has some limitations. First, although we demonstrated that mobilization of Scx-GFP+ cells into a reconstructed ligament was more efficient in the presence of remnant tissue, this study was performed in rats, and it is unclear whether these findings are directly applicable to humans. Second, the function of Scx-expressing cells that mobilize to a reconstructed ligament is not completely clear and requires further analysis. Nonetheless, we feel that our demonstration that mobilization of Scx-positive cells to a reconstructed ligament likely increases initial strength of a reconstructed ligament is of significant value to the field.

To date, there have been various attempts to decrease the postoperative rest period and expedite the return to sports following ACL reconstruction surgery, among them, application of a cell sheet generated by cultured mesenchymal stem cells to the graft tendon, as performed in rabbits [28]. It is also reported that Scx expression is upregulated by various factors including TGF-β or FGF [27]. However, to date, no method to speed recovery has been applied clinically. Based on our findings, monitoring Scx expression in a model comparable to our Scx-GFP transgenic rats could serve as a means to identify additional agents to enhance maturation and strength of grafted tendons after ACL reconstruction surgery. Moreover, our model provides a valuable tool to monitor Scx-positive cell infiltration of grafted tendons after surgery *in vivo*.

## Supporting information

**S1 Fig. The ACL of 1-day-old Scx-GFP Tg rats exhibits Scx-GFP-expressing cells.** Knee joint tissues were collected from 1-day-old Scx-GFP Tg rats and stained with anti-GFP antibody, as a means to track Scx-expressing cells based on GFP fluorescence. Nuclei are DAPI-stained. Magnifications are as indicated. Bar, 100μm.
(TIF)

**S2 Fig. Schematic describing creation of ACL reconstruction models.** In remnant preservation models (Group P), the ACL was dissected at the center of the parenchyma, bone tunnels were created in the vicinity of the dissected ACL, and the graft tendon was inserted. In remnant resection models (Group R), the ACL was dissected from bony attachments, and the graft tendon was inserted at the anatomic location of the ACL. In addition, for ACL tear models, the ACL was dissected at the center of the parenchyma, and closed wound models were used.
(TIF)

**S3 Fig. Schematic showing rat sample sizes and experimental design of biomechanical strength analysis.** ACL reconstruction models were divided into remnant preservation models and resection models. For both, knee joint tissues from wild-type male rats were collected at 4, 6, or 8 weeks postoperatively and subjected to tensile tear testing. As indicated, sample size in all groups was 14 rats per group.
(TIF)

**S4 Fig. Schematic showing ACL reconstruction models based on allogeneic tendon transplantation.** To determine the origin of Scx-expressing cells, three types of ACL allogeneic reconstruction models were created using 11 to 12-week-old Scx-GFP Tg or wild-type male rats. In the first, wild-type rats are transplanted with grafts from wild-type rats. In the second, Scx-GFP Tg rats are transplanted with grafts from wild-type rats, and in the third, wild-type

rats are transplanted with grafts from Scx-GFP Tg rats.
(TIF)

**S5 Fig. GFP-positive cells are absent in the flexor digitorum longus tendon derived from adult Scx-GFP Tg rats prior to transplantation.** Long digitorum flexor tendons from 11 to 12-week-old Scx-GFP Tg rats were harvested, paraformaldehyde-fixed, freeze-embedded, and sectioned (5μm). Sections were then stained with anti-GFP antibody to detect Scx-expressing cells, as reflected by GFP positivity, within the grafted tendon prior to implantation. Nuclei are DAPI-stained. Note that no GFP-positive cells are present in the flexor digitorum longus tendon prior to transplant. As a positive control, we used Achilles tendons from 1-day-old Scx-GFP Tg rats. Magnifications are as indicated. Bar, 100μm.
(TIF)

**S1 File. Raw data for mechanical evaluation, histological evaluation and RT-PCR.**
(DOCX)

## Author Contributions

**Conceptualization:** Tetsuro Masuda.

**Data curation:** Tetsuro Masuda.

**Formal analysis:** Junki Kawakami.

**Funding acquisition:** Tetsuro Masuda.

**Methodology:** Satoshi Hisanaga, Yuki Yoshimoto, Tomoji Mashimo, Takehito Kaneko, Naoto Yoshimura, Masaki Shimada, Makoto Tateyama, Hideto Matsunaga, Yuto Shibata, Shuntaro Tanimura, Kosei Takata, Takahiro Arima, Kazuya Maeda, Yuko Fukuma, Masaru Uragami, Katsumasa Ideo, Kazuki Sugimoto, Ryuji Yonemitsu, Kozo Matsushita, Masaki Yugami, Yusuke Uehara, Takayuki Nakamura, Takuya Tokunaga, Tatsuki Karasugi, Takanao Sueyoshi, Chisa Shukunami, Nobukazu Okamoto, Tetsuro Masuda.

**Project administration:** Tetsuro Masuda.

**Supervision:** Tetsuro Masuda, Takeshi Miyamoto.

**Validation:** Junki Kawakami.

**Visualization:** Junki Kawakami.

**Writing – original draft:** Junki Kawakami.

**Writing – review & editing:** Tetsuro Masuda.

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
