## [Decision Letter · Decision Letter 0]

25 Aug 2023

PONE-D-23-22734Remnant tissue enhances early postoperative biomechanical strength and infiltration of Scleraxis-positive cells within the grafted tendon in a rat anterior cruciate ligament reconstruction modelPLOS ONE

Dear Dr. Miyamoto,

Thank you for submitting your manuscript to PLOS ONE. After careful consideration, we feel that it has merit but does not fully meet PLOS ONE’s publication criteria as it currently stands. Therefore, we invite you to submit a revised version of the manuscript that addresses the points raised during the review process.

We look forward to receiving your revised manuscript.

Kind regards,

Charles Neil Pagel

Academic Editor

PLOS ONE

Journal Requirements:

2. To comply with PLOS ONE submissions requirements, in your Methods section, please provide additional information regarding the experiments involving animals and ensure you have included details on (1) methods of sacrifice,, and (2) efforts to alleviate suffering.

Dear Dr Miyamoto,

Thank you for submitting your manuscript ID PONE-D-23-22734 "Remnant tissue enhances early postoperative biomechanical strength and infiltration of Scleraxis-positive cells within the grafted tendon in a rat anterior cruciate ligament reconstruction model" to PLOS One.

The manuscript has been reviewed by two expert referees. We require major revisions to your manuscript before it can be reconsidered for publication.

Therefore, I invite you to respond to the comments included at the bottom of this letter and revise your manuscript accordingly.

Yours sincerely

Charles Pagel

Academic Editor

Reviewers' comments:

Reviewer's Responses to Questions

**Comments to the Author**

1. Is the manuscript technically sound, and do the data support the conclusions?

Reviewer #1: Yes

Reviewer #2: No

2. Has the statistical analysis been performed appropriately and rigorously? 

Reviewer #1: Yes

Reviewer #2: I Don't Know

3. Have the authors made all data underlying the findings in their manuscript fully available?

Reviewer #1: Yes

Reviewer #2: No

4. Is the manuscript presented in an intelligible fashion and written in standard English?

Reviewer #1: Yes

Reviewer #2: No

5. Review Comments to the Author

Reviewer 1

I read your paper titled "Remnant tissue enhances early postoperative biomechanical strength and infiltration of Scleraxis-positive cells within the grafted tendon in a rat anterior cruciate ligament reconstruction model" with great interest.

The topic of your research is truly fascinating. However, I would like to suggest some improvements that could enhance the overall quality of your paper. Firstly, it would be beneficial to include a section discussing the limitations of your study. This would provide readers with a more comprehensive understanding of the scope and potential areas for future research.

Additionally, I believe it would be valuable to expand upon the clinical implications of your findings. How could the results of your study be applied in real-world scenarios or medical practices? This would help bridge the gap between your research and its practical implications.

Lastly, I noticed that the specific contributions of each author were not provided in the paper. It would be beneficial to clarify the roles of all the authors in the research to acknowledge their respective contributions appropriately.

Overall, your paper presents valuable insights into the subject matter, and with these suggested improvements, it has the potential to become an even more impactful piece of research.

Reviewer 2

I appreciate the efforts put forth by the authors on this study. I have the following comments based on my understanding of the manuscript as presented.

Abstract

1) The abstract likely needs to be re-structured for the readership. The abstract should have a concise reason for the study, overall study objective(s), and hypothesis. It should then have a concise research method including animal species, numbers, and model used. This is then followed by a concise review of the study outcomes including numerical results. It then concludes with the important implications of the study findings and what it adds to the literature.

2) Can the authors clarify why FDP tendons used for ACL reconstructions would not have Scx producing cells as mentioned in the abstract. Did the authors do several analyses that could confirm this fact (lack of scleraxis cells in the FDP tendons) prior to implanting the graft? Scx is involved in a variety of tendon/ligament injury and healing processes so I am not clear why Scx would be absent in the tendons. I can potentially understand if the authors used wild type FDL tendons to place into GFP Scx rats, which then would allow them to potentially see where the GFP Scx cells would be in wild-type FDLs used for ACL reconstructions.

Introduction

3) While double bundle reconstruction techniques are one option for ACL reconstruction, the authors may want to amend lines 65-66 to include both single and double bundle ACL reconstruction techniques.

4) Lines 68-71 should be rewritten as the thoughts presented is not clear. I believe the authors are trying to convey that the ACL graft used undergo ligamentization which includes graft necrosis followed by cellular infiltration and remodeling of the placed ACL graft. This may compromise the strength of the ACL graft.

5) “Maximum breaking load” in line 78 should be changed to “load-to-failure”

6) “Residual tissue” in line 82 should likely be corrected to “native ACL remnant tissue”

7) Line 85 should be corrected to "which may facilitate healing of the transplanted tendon”

8) The authors may consider adding their study hypothesis in the last paragraph of the introduction

Methods

9) The authors should consider adding the age, sex, weight of the rats used in the experiment. They should also include the number of rats used for each group and also the sample size for each type of analysis including biomechanical, histology, etc. They should consider constructing a figure that lays out the experiment groups and analyses along with sample sizes.

10) I did not have a clear understanding how the remnant preserving and resections groups were created based on the methods section. This was not described well in the research methods in terms of where these two groups were created and how they differed. It was also not clear if WT grafts were placed in GFP rats and vice versa based on the methods. It appears based on the results that this may have occurred. A figure showing the different surgery groups (remnant preservation vs. resection) and how they were created would be helpful as well so one can determine how those conditions were created.

11) Line 123 should likely be “A medial parapatellar approach was used to reach the joint cavity…”

12) Line 124-125 should likely read “The ACL was identified and an ACL tear was created by cutting the central portion of the ACL.”

13) “Foramen” in line 135 should likely be “tunnel”

14) Can the authors clarify where the FDL grafts came from for WT and GFP rats undergoing ACL reconstructions? Was there a situation where WT FDL tendons were placed in GFP rats? I think the results section suggests this happened but this was not described in detail in the methods sections.

15) Can the authors clarify the rationale behind having an ACL tear model versus a reconstruction model for the purposes of this experiment? If the goals are to quantify the contribution of Scx expressing cells after ACL reconstruction, I did not understand the use of an ACL tear group alone without reconstruction.

16) Can the authors clarify if the nylon sutures that were used to fix the ACL graft both on the femur and tibia were removed prior biomechanical testing? This should be clarified in the description of the study. If it was not released, can the authors clarify why not?

17) The authors should consider providing better description of the immunohistochemical analyses. These were 5 micrometer sections. Which tissues were analyzed (ACL graft, ACL remnant)? Can the authors also clarify why there were different time points for different tissues (Lines 162 and 164)? How many sections were used for analyses and how were the analyses combined for different sections? Was the entire tissue analyzed or were there specific regions that were selected for semiquantitative analyses? What magnifications were used for the analyses?

18) Can the authors explain why the did not do histological analyses at all of the time points as the biomechanical analyses considering the goal was to correlate the number of GFP Scx cells and see if it impacted biomechanical strength (i.e., 8 weeks)? This also goes for the RT-PCR analyses. I felt this lack of congruity was a significant weakness as the objective was to correlate GFP cells and biomechanical strength of the ACL grafts.

19) The authors should include a statistical analysis section. It would be ideal that they detail the statistical methods used for their study analyses, the significance level used, and a power analysis if it was done to justify sample size.

Result

20) Can the authors discuss where the failure site was for each group during load-to-failure testing? Was it graft tunnel pullout or mid-graft? Was it femoral sided or tibial sided? They should include this data as it provides a clue where the weak-link is at each time point for the load-to-failure testing. This would also allow one to judge if GFP in the ACL graft substance matter. Was stiffness calculated for the load-to-failure testing as well?

21) The result section, in particular the histological analysis section, should be better structured. While qualitative statement that one group is better than others was included in the results section, there were no quantitative numbers provided including p-values to justify those statements.

22) The RT-PCR data was not presented in written form in the result section. As this is part of the methods, the authors should include a written explanation of the results.

23) Can the authors clarify how some of the biomechanical results were statistically significant given some of the large and overlapping standard deviations (Figure 1)? This highlights the need for that authors to provide better description of groups sizes and statistical tests used.

Discussion

24) I think a part of the issue with the study is that the timepoints for histological analyses did not correspond with the biomechanical analyses. Therefore, the link between GFP cell numbers and biomechanical strength is not as direct as the authors suggest (lines 346-349). This is particularly true for latter time points (8-week) where the differences between remnant preservation and resection was not present.

25) The authors may want to reconsider their discussion section in terms of salient points to discuss in order to highlight the importance of their paper and its results. I think the authors should highlight what is known about Scx and ACL graft maturation. They should also discuss how their use of GFP-labeled Scx cells provide an understanding of how graft ligamentization occur in terms of cellular necrosis, followed by cellular invasion, and ligamentization.

Grammatical/Formatting Suggestions

26) The manuscript would benefit from an editor to adjust/correct for the English language and grammatical conventions.

Figures and Tables

26) The histological images would benefit from having magnification and scale bars.

6. PLOS authors have the option to publish the peer review history of their article (what does this mean?). If published, this will include your full peer review and any attached files.

Reviewer #1: No

Reviewer #2: No

---

## [Author Response · Author response to Decision Letter 0]

4 Oct 2023

Response to Reviewers

Responses to Reviewer 1

I read your paper titled "Remnant tissue enhances early postoperative biomechanical strength and infiltration of Scleraxis-positive cells within the grafted tendon in a rat anterior cruciate ligament reconstruction model" with great interest.

Reply: Thank you for your positive comments on our manuscript.

The topic of your research is truly fascinating. However, I would like to suggest some improvements that could enhance the overall quality of your paper. Firstly, it would be beneficial to include a section discussing the limitations of your study. This would provide readers with a more comprehensive understanding of the scope and potential areas for future research.

Reply: In response to this suggestion we now discuss study limitations near the end of the Discussion section (lines 465-472). In brief, we note that this study was performed in rats, and thus it remains unknown whether our findings will apply directly to humans. Second, we state that further studies are needed to determine the function of Scx-positive cells in re-ligamentation and in increasing biomechanical strength of grafted tendons. 

Additionally, I believe it would be valuable to expand upon the clinical implications of your findings. How could the results of your study be applied in real-world scenarios or medical practices? This would help bridge the gap between your research and its practical implications.

Reply: To date, various attempts have been made to increase biomechanical strength of grafted tendons and shorten the rest period after surgery. However, no methods to achieve these goals have yet been applied clinically. Based on our findings, we conclude monitoring changes in Scx expression could serve as a tool to identify additional agents that may enhance maturation and strength of grafted tendons after ACL reconstruction surgery. We now state this in the Discussion section (lines 473-482). 

Lastly, I noticed that the specific contributions of each author were not provided in the paper. It would be beneficial to clarify the roles of all the authors in the research to acknowledge their respective contributions appropriately.

Reply: Specific author contributions are now found in “Author Contributions” at the end of the paper on lines 489-495.

Overall, your paper presents valuable insights into the subject matter, and with these suggested improvements, it has the potential to become an even more impactful piece of research.

Reply: We thank the reviewer for these positive comments.

 

Responses to Reviewer 2

I appreciate the efforts put forth by the authors on this study. I have the following comments based on my understanding of the manuscript as presented.

Abstract

1) The abstract likely needs to be re-structured for the readership. The abstract should have a concise reason for the study, overall study objective(s), and hypothesis. It should then have a concise research method including animal species, numbers, and model used. This is then followed by a concise review of the study outcomes including numerical results. It then concludes with the important implications of the study findings and what it adds to the literature.

Reply: Accordingly, we completely re-structured the Abstract, as requested, to emphasize the study’s rationale, its objective(s), our hypothesis and methods. In the revised Abstract, we also now briefly summarize outcomes and the significance of this work to the field.

2) Can the authors clarify why FDP tendons used for ACL reconstructions would not have Scx producing cells as mentioned in the abstract. Did the authors do several analyses that could confirm this fact (lack of scleraxis cells in the FDP tendons) prior to implanting the graft? Scx is involved in a variety of tendon/ligament injury and healing processes so I am not clear why Scx would be absent in the tendons. I can potentially understand if the authors used wild type FDL tendons to place into GFP Scx rats, which then would allow them to potentially see where the GFP Scx cells would be in wild-type FDLs used for ACL reconstructions.

Reply: We have now confirmed that prior to graft implantation, FDP tendons derived from Scx-GFP Tg rats do not exhibit GFP-positive cells. This data is shown in new Figure S5 and reported in the Results on lines 287-288. However, when we transplanted FDL tendons from wild type rats into GFP-Scx rats, we detected GFP+ cells in grafted tendons. By contrast, when we transplanted tendons from adult Scx-GFP+ rats into wild type rats, no GFP Scx cells were detected in the grafted tendon. These data are now shown in Figure 4 and reported in the Results on lines 341-349.

At present, we do not know why adult tendons derived from Scx-GFP Tg rats do not exhibit GFP positivity. However, these findings may be analogous to other systems in which embryonic or developmental gene expression programs re-emerge in adult animals following injury. For example, chondrocytes are required for bone growth during endochondral ossification, which occurs at embryonic and developmental stages, but are not present in adults [22]. However, during healing of a fracture, the developmental endochondral ossification program re-emerges [23]. We conclude that Scx-positive cells likely contribute to development and subsequent healing of tendons and ligaments. We now discuss these points in the Discussion on lines 412-435.

Introduction

3) While double bundle reconstruction techniques are one option for ACL reconstruction, the authors may want to amend lines 65-66 to include both single and double bundle ACL reconstruction techniques.

Reply: We amended this passage (now, lines 65-66) to include both single- and double-bundle ACL reconstruction techniques.

4) Lines 68-71 should be rewritten as the thoughts presented is not clear. I believe the authors are trying to convey that the ACL graft used undergo ligamentization which includes graft necrosis followed by cellular infiltration and remodeling of the placed ACL graft. This may compromise the strength of the ACL graft.

Reply: We appreciate this suggestion and have rewritten this passage accordingly in the revision. 

5) “Maximum breaking load” in line 78 should be changed to “load-to-failure”

Reply: As suggested, “Maximum breaking load” was changed to “load-to-failure”.

6) “Residual tissue” in line 82 should likely be corrected to “native ACL remnant tissue”.

Reply: Accordingly, we changed “Residual tissue” to “native ACL remnant tissue”.

7) Line 85 should be corrected to "which may facilitate healing of the transplanted tendon”

Reply: Accordingly, this passage (line 82 in the revision) has been changed to "which may facilitate healing of the transplanted tendon”. 

8) The authors may consider adding their study hypothesis in the last paragraph of the introduction

Reply: We now state the study’s hypothesis in the last paragraph of the Introduction on lines 93-95, as requested.

Methods

9) The authors should consider adding the age, sex, weight of the rats used in the experiment. They should also include the number of rats used for each group and also the sample size for each type of analysis including biomechanical, histology, etc. They should consider constructing a figure that lays out the experiment groups and analyses along with sample sizes.

Reply: In the Materials and Methods section, we now state that we used 11 to 12-week-old WT or Tg male rats (body weight, 424.2 ± 17.4 g) to create animal models of ACL reconstruction in a remnant preservation group (Group P) and remnant resection group (Group R). We also now report rat body weight at the time of biomechanical testing. This information is reported on lines 150-153. Finally, we now include sample size for each analysis in relevant figure legends and have constructed a new figure (Figure S3) that illustrates experimental groups and states sample sizes.

10) I did not have a clear understanding how the remnant preserving and resections groups were created based on the methods section. This was not described well in the research methods in terms of where these two groups were created and how they differed. It was also not clear if WT grafts were placed in GFP rats and vice versa based on the methods. It appears based on the results that this may have occurred. A figure showing the different surgery groups (remnant preservation vs. resection) and how they were created would be helpful as well so one can determine how those conditions were created.

Reply: Accordingly, we provided a new figure (Figure S2) to show different surgery groups (remnant preservation vs. resection) and how they were created. We also prepared a new figure (Figure S4) showing how wild type grafts were placed in Scx-GFP rats, and vice versa, and describe these procedures in the Materials and Methods section on lines 176-182.

11) Line 123 should likely be “A medial parapatellar approach was used to reach the joint cavity…”

Reply: We have now made this precise change on line 125, and thank you for the suggestion.

12) Line 124-125 should likely read “The ACL was identified and an ACL tear was created by cutting the central portion of the ACL.”

Reply: Accordingly, lines 126-127 were changed to read “The ACL was identified and an ACL tear was created by cutting the central portion.”

13) “Foramen” in line 135 should likely be “tunnel”

Reply: We replaced “Foramen” on line 137 with “tunnel”.

14) Can the authors clarify where the FDL grafts came from for WT and GFP rats undergoing ACL reconstructions? Was there a situation where WT FDL tendons were placed in GFP rats? I think the results section suggests this happened but this was not described in detail in the methods sections.

Reply: In the revised Methods section (lines 176-182), we now state what animals provided the FDL grafts and which rats those grafts were transplanted into—i.e., either wild type or Scx-GFP. We also added a new figure (Figure S4) to clearly show methods used for ACL reconstructions.

15) Can the authors clarify the rationale behind having an ACL tear model versus a reconstruction model for the purposes of this experiment? If the goals are to quantify the contribution of Scx expressing cells after ACL reconstruction, I did not understand the use of an ACL tear group alone without reconstruction.

Reply: We chose to employ a ACL tear model alone without reconstruction because we hypothesized that remnant tissue might be a source of Scx-positive cells. We believed that a tear model might be a more appropriate system in which to test this hypothesis. 

16) Can the authors clarify if the nylon sutures that were used to fix the ACL graft both on the femur and tibia were removed prior biomechanical testing? This should be clarified in the description of the study. If it was not released, can the authors clarify why not?

Reply: Nylon sutures used to fix the graft both on the femur and tibia were removed prior biomechanical testing, as now stated explicitly in the Materials and Methods section in lines 160-161.

17) The authors should consider providing better description of the immunohistochemical analyses. These were 5 micrometer sections. Which tissues were analyzed (ACL graft, ACL remnant)? Can the authors also clarify why there were different time points for different tissues (Lines 162 and 164)? How many sections were used for analyses and how were the analyses combined for different sections? Was the entire tissue analyzed or were there specific regions that were selected for semiquantitative analyses? What magnifications were used for the analyses?

Reply: As requested, we now provide more concise descriptions of immunohistochemical analyses performed in grafted tendons and remnant ACL models tested. These points are now more clearly stated in various parts of the Materials and Methods section, including on lines171-176 and on lines 196-209.

In brief, we performed immunohistochemical staining of ACL remnant tissue up to 6 weeks after the ACL was cut. We were initially interested primarily in the 4 week time point but decided to assess potential changes also at 6 weeks. To do so, we used one section from each experimental sample, each representing the ACL reconstruction model, the parenchyma of the ACL graft tendon outside the bony tunnel and the ACL remnant for the tear model. In all analyses, tissue images at 40x magnification were evaluated. 

18) Can the authors explain why the did not do histological analyses at all of the time points as the biomechanical analyses considering the goal was to correlate the number of GFP Scx cells and see if it impacted biomechanical strength (i.e., 8 weeks)? This also goes for the RT-PCR analyses. I felt this lack of congruity was a significant weakness as the objective was to correlate GFP cells and biomechanical strength of the ACL grafts.

Reply: We apologize for the apparent “lack of congruity” in performing histological and biochemical analyses. In short, we assessed biomechanical strength of a transplant, as well as Scx mRNA and protein levels, at what we judged were critical time points. For example, biomechanical strength was significantly higher in the groups P than R at 4 and 6 weeks post-surgery, but was comparable in both groups at 8 weeks. By contrast, the area of GFP-positive cells was significantly larger in group P than R at 4 weeks but those areas were comparable at 6 weeks. Also, Scx transcript expression plateau’d by 2 weeks post-surgery. We concluded that increases in Scx mRNA and protein occur prior to increases in transplant strength. Thus, we assessed biomechanical strength at time points after upregulation of Scx mRNA and protein.

19) The authors should include a statistical analysis section. It would be ideal that they detail the statistical methods used for their study analyses, the significance level used, and a power analysis if it was done to justify sample size.

Reply: As requested, now we include a statistical analysis section in the Materials and Methods section.

Result

20) Can the authors discuss where the failure site was for each group during load-to-failure testing? Was it graft tunnel pullout or mid-graft? Was it femoral sided or tibial sided? They should include this data as it provides a clue where the weak-link is at each time point for the load-to-failure testing. This would also allow one to judge if GFP in the ACL graft substance matter. Was stiffness calculated for the load-to-failure testing as well?

Reply: We found that no tendon could be pulled out through the bone tunnel at any time point after surgery. Instead, in all cases, grafted tendons ruptured in the center of the parenchyma. We now state this clearly in the Results on lines 245-247. Note that Scx-GFP rats were not used for biomechanical testing: those analyses were performed in WT rats, as illustrated in new Figure S3. Finally, we calculated stiffness for load-to-failure testing as well, and that data is shown in new Figure 1C and discussed in the Results on lines 258-261.

21) The result section, in particular the histological analysis section, should be better structured. While qualitative statement that one group is better than others was included in the results section, there were no quantitative numbers provided including p-values to justify those statements.

Reply: As requested, we now report p-values for each experiment in the Results section. We also show p-values in the Figures and corresponding legends.

22) The RT-PCR data was not presented in written form in the result section. As this is part of the methods, the authors should include a written explanation of the results.

Reply: We now present RT-PCR data in the Results section narrative on lines 324-326.

23) Can the authors clarify how some of the biomechanical results were statistically significant given some of the large and overlapping standard deviations (Figure 1)? This highlights the need for that authors to provide better description of groups sizes and statistical tests used.

Reply: In the revision we show p-values in figures and corresponding legends and also define sample size in each legend. To statistically analyze data shown in Fig. 1, we performed Student’s t-test or the Mann-Whitney U test. Statistical analysis of 3 or more subgroups was performed using one-way ANOVA, followed by a Tukey–Kramer test. Those tests confirmed that differences observed—despite the error—were significant, and we stand by those conclusions. 

Discussion

24) I think a part of the issue with the study is that the timepoints for histological analyses did not correspond with the biomechanical analyses. Therefore, the link between GFP cell numbers and biomechanical strength is not as direct as the authors suggest (lines 346-349). This is particularly true for latter time points (8-week) where the differences between remnant preservation and resection was not present.

Reply: As stated in the manuscript and in our reply to your question #18, we conclude that the transplanted tendon matures and exhibits increases in the biomechanical strength over time in this experimental model, and that this occurs after upregulation of Scx mRNA and protein. Thus, as the reviewer notes, there is a time lag between appearance of GFP+ cells and increases in biomechanical strength, which are likely dependent on maturation of those GFP+ cells into tenocytes. We now discuss these points in the Discussion on lines 375-401. 

25) The authors may want to reconsider their discussion section in terms of salient points to discuss in order to highlight the importance of their paper and its results. I think the authors should highlight what is known about Scx and ACL graft maturation. They should also discuss how their use of GFP-labeled Scx cells provide an understanding of how graft ligamentization occur in terms of cellular necrosis, followed by cellular invasion, and ligamentization.

Reply: First, we thank the Reviewer for suggesting ways to restructure our Discussion to more clearly highlight salient results and summarize why they are important. Accordingly, we restructured the Discussion in several ways. Briefly, first, we highlighted what is known about Scx and ACL graft maturation. Scx is known to be required for embryonic and post-natal growth of ACL and other tendons and ligaments, but its role in tendon repair in adults is unknown. These points are now discussed in the Discussion on lines 380-381. Then, here we established Scx-GFP rats and observed that Scx-GFP+ cells, although absent in mature adult tendons, appeared in the graft from outside the tendon during re-ligamentization after ACL reconstruction surgery. These findings are novel, and were not previously described. We now discuss these points in the Discussion on lines 381-384.

Grammatical/Formatting Suggestions

26) The manuscript would benefit from an editor to adjust/correct for the English language and grammatical conventions.

Reply: This revision has been edited by a native English-speaking scientist.

Figures and Tables

26) The histological images would benefit from having magnification and scale bars.

Reply: Magnification of histological images is now stated in relevant figure legends.

---

## [Decision Letter · Decision Letter 1]

23 Oct 2023

Remnant tissue enhances early postoperative biomechanical strength and infiltration of Scleraxis-positive cells within the grafted tendon in a rat anterior cruciate ligament reconstruction model

PONE-D-23-22734R1

Dear Dr. Miyamoto,

We’re pleased to inform you that your manuscript has been judged scientifically suitable for publication and will be formally accepted for publication once it meets all outstanding technical requirements.

Kind regards,

Charles Neil Pagel

Academic Editor

PLOS ONE

Additional Editor Comments (optional):

Reviewers' comments:

Reviewer's Responses to Questions

**Comments to the Author**

1. If the authors have adequately addressed your comments raised in a previous round of review and you feel that this manuscript is now acceptable for publication, you may indicate that here to bypass the “Comments to the Author” section, enter your conflict of interest statement in the “Confidential to Editor” section, and submit your "Accept" recommendation.

Reviewer #1: All comments have been addressed

2. Is the manuscript technically sound, and do the data support the conclusions?

Reviewer #1: Yes

3. Has the statistical analysis been performed appropriately and rigorously? 

Reviewer #1: I Don't Know

4. Have the authors made all data underlying the findings in their manuscript fully available?

Reviewer #1: No

5. Is the manuscript presented in an intelligible fashion and written in standard English?

Reviewer #1: Yes

6. Review Comments to the Author

Reviewer #1: Dear authors,

The paper is well written and you have addressed all my queries. I recommend acceptance.

7. PLOS authors have the option to publish the peer review history of their article (what does this mean?). If published, this will include your full peer review and any attached files.

Reviewer #1: No

---

## [Editor Report · Acceptance letter]

31 Oct 2023

PONE-D-23-22734R1 

Remnant tissue enhances early postoperative biomechanical strength and infiltration of Scleraxis-positive cells within the grafted tendon in a rat anterior cruciate ligament reconstruction model 

Dear Dr. Miyamoto:

I'm pleased to inform you that your manuscript has been deemed suitable for publication in PLOS ONE. Congratulations! Your manuscript is now with our production department. 

Kind regards, 

on behalf of

Dr. Charles Neil Pagel 

Academic Editor

PLOS ONE